# Do We Learn to Internalize Stigma from Our Parents? Comparison of Internalized Stigmatization in Adolescents Diagnosed with ADHD and Their Parents

**Gül Dikeç** [1,*] **, Öznur Bilaç** [2] **, Cansın Kardelen** [2] **and Şermin Yalin Sapmaz** [2]

1. Department of Nursing, Faculty of Health Sciences, Fenerbahce University, Istanbul 34758, Turkey
2. Department of Child and Adolescent Psychiatry, Hafsa Sultan Medical School Hospital, Celal Bayar University, Manisa 45030, Turkey
* Correspondence: gul.dikec@fbu.edu.tr; Tel.: +90-216-910-19-07

**Abstract:** This study compared internalized stigmatization levels of adolescents diagnosed with attention deficit and hyperactivity disorder (ADHD) with those of their parents. The study's data were collected from 107 adolescents diagnosed with ADHD and their parents between July 2020 and March 2021. The adolescents were followed up in the child and adolescent psychiatry outpatient clinic of a university hospital in western Turkey. The information forms for adolescents and parents, the Internalized Stigma of Mental Illness Scale—Adolescent Form (ISMI-AF) and the Parental Internalized Stigma of Mental Illness Scale (PISMI), were used to collect the data. There was no statistically significant difference between the total scores of internalized stigma and subscale mean scores of the adolescents and their parents ($p > 0.05$); only the subscale scores for stereotype endorsement were found to be significantly different ($p < 0.05$). PISMI scores affected ISMI-AF scores, which can be interpreted as parents' perspectives and attitudes toward stigmatization affecting adolescents. For ADHD, whose frequency is increasing daily, intervention studies should be conducted to reduce adolescents' and parents' internalized stigma and to enhance the educational outcomes of adolescents.

**Keywords:** adolescents; adolescent psychiatry; ADHD; stigmatization

## 1. Introduction

Attention deficit hyperactivity disorder (ADHD) is characterized by problems and behaviors that are inappropriate to age and developmental stages, such as attention problems, forgetfulness, hyperactivity, and impulsivity. According to DSM-V diagnostic criteria, ADHD is defined by the presence of a persistent pattern of attention deficit and hyperactivity–impulsivity that begins before the age of 12 and impairs functionality and development, and these symptoms must last for at least six months to be diagnosed. It is a neuropsychiatric disorder that starts early in life, and its effects continue throughout life [1]. The prevalence of ADHD was found to be 3.4% in a systematic meta-analysis study that evaluated worldwide studies [2]. The EPICPAC-T (Epidemiology of Childhood Psychopathology in Turkey), the largest epidemiologic face-to-face survey of school-age children in Turkey, aimed to determine the prevalence of childhood psychopathologies. According to this study, ADHD is the most common disorder with (19.5%) and without impairment (16.7%) among 6- to 13-year-old school children (n = 5830) [3].

The World Health Organization [4] has stated that the most critical barrier that individuals meet when they receive initial treatment for mental health is stigma and discrimination. Stigma can be defined in three ways: public stigma, self-stigma or internalized stigma, and courtesy stigma [5]. Public stigma is characterized by the negative attitudes and behaviors of society toward individuals with mental illnesses. Internalized stigma means that after being diagnosed with a mental illness, the individual internalizes negative attitudes toward mental illnesses in society [6–8]. In addition, courtesy stigma, defined by Goffman [9],

represents the phenomenon that family members or people close to a stigmatized person are negatively judged due to their mere association with the stigmatized target [10]. Public stigma, internalized stigma, and courtesy stigma are phenomena that people with mental illnesses and their family members can frequently face and feel in their daily lives. These phenomena affect self-esteem and patient outcomes, such as medication adherence, quality of life, and functioning of people with mental illnesses, as well as the quality of life of their family members and people close to them [5,7,10].

It is thought that some children and adolescents diagnosed with ADHD are unpredictable, unreliable, impolite, stressed, weak, and immature. Therefore, people could stigmatize and discriminate against children and adolescents with ADHD in social and school environments [7]. In addition to the problems in their daily lives [11], they experience public stigma in school and social environments [12]. Mueller et al. [10] have explained that children and adolescents with ADHD are stigmatized by peers, teachers, neighbors, family members, and by themselves. Some parents expressed a strong concern about having a child with ADHD move in next door; they were also unwilling to allow their child to befriend or to be in the same classroom as a child with ADHD. At school, children and adolescents also experienced stigma, in that they are perceived by their teachers to perform significantly worse in both math and reading than their classmates without an ADHD diagnosis. In addition, parents' attitudes are similar to those of teachers and adolescents who harbor negative attitudes [13]. They may internalize this public stigma [12]. Moses [14] found that among adolescents, there was a strong correlation between self-stigma and the perception of having social skills deficits. Internalized stigma has important effects on the personality development of adolescents, the severity of the symptoms, their enjoyment of life, and the mental well-being of individuals with ADHD [15]. However, there are a limited number of studies that have evaluated internalized stigmatization. McKeague et al. [15] examined retrospective accounts of self-stigma experienced by young people with ADHD and found that young people were aware of being stigmatized by their peers and felt different from others, in addition to experiencing personal distress, self-devaluation, and self-blame. This study demonstrates that serious negative consequences are associated with internalized stigma.

The public also stigmatizes the parents of these adolescents, hence parents experience "courtesy stigma" [12]. Parents are accused of poor parenting [16] and experience unpleasant emotions such as anger, guilt, and shame [9]. Parents might feel guilty and embarrassed because their child's illnesses are inherited from them or because they believe that they cannot be good parents or protect them [5,8]. Therefore, they are vulnerable to experiencing courtesy stigma and internalize negative commentary, and feedback can precipitate increased parenting stress [13]. Furthermore, they often suffer parental stress that can affect the outcomes of their children [9]. Charboiner et al. [16] found that the internalized stigmatization of mothers of children with ADHD was an important factor in parental stress, and the authors emphasized that internalized stigma is a strong variable to take into account in the clinical management of ADHD. Since parents may think that being diagnosed and getting treatment make stigmatization worse for their children, they may prefer for their children not to initiate or maintain the treatment as prescribed, to protect them from public stigma [11,17]. Therefore, the internalized stigmatization of the parents could influence their children and negatively affect their attitudes toward diagnosis and treatment [18,19].

It is known that children's attitudes are shaped at early times, continuing into adulthood, and that cultural attributions play a critical role in shaping attitudes toward mental disorders [20–22]. Parents are the most potent source of the attitudes of children, and most children have the same attitudes as their parents or are affected by their perspectives. Elkington et al. [22] found that adolescents with mental illnesses were first stigmatized within their family and their close social circle. Therefore, it is essential to examine the attitudes of children and their parents.

As far as we know, there had been no quantitative study to evaluate and compare internalized stigmatization in both adolescents with ADHD and their parents. Therefore, this study aimed to compare the internalized stigmatization levels of both groups.

## 2. Materials and Methods

The Strengthening the Reporting of Observational Studies in Epidemiology (STROBE) Checklist [23], which is commonly used in the standardization of descriptive and observational studies, was used to report the present study.

### 2.1. Study Design

This is a descriptive cross-sectional study.

### 2.2. Data Collection Procedure

The study data were collected in the child and adolescent psychiatry outpatient clinic of a university hospital in western Turkey between July 2020 and March 2021, during the COVID-19 pandemic. When the adolescents applied for an appointment, the primarily responsible child and adolescent psychiatrists (Ö.B. and Ş.Y.S.) invited the adolescents and their parents to participate in the study. The adolescents with ADHD and the parents who agreed to participate voluntarily were invited to an empty room to complete the forms individually. The parents who accompanied their children were invited to participate in the study when they arrived at the hospital. No couples accompanied the adolescents; only one parent of each adolescent was included in the study. Adolescents who applied to the child and adolescent outpatient clinic without their parents were excluded.

### 2.3. Participants

The study consisted of adolescents, aged 12–18 years, who agreed to participate in the research and who had a diagnosis of ADHD according to the DSM-V. Between 2019 and 2020, a total of 1610 children and adolescents with various mental problems and 215 children and adolescents with ADHD attended the outpatient clinic where the data were collected. Due to the decrease in the number of patients who applied to outpatient clinics because of the pandemic, 114 adolescents were admitted to the clinic in the data collection period, and 7 of them refused to participate. A total of 107 adolescents and their parents agreed to participate in the study. The sample of the study consisted of adolescents who applied to the outpatient clinic with their parents at the time of the study, met the sample selection criteria, and parents between the ages of 18 and 65 years who did not have any mental illness and agreed to participate in the study. Adolescents with comorbid illnesses were excluded from the study. A total of 107 adolescents and 107 parents took part in the study.

### 2.4. Measurements

In this study, two information forms were used to collect data on adolescents' and their parents' socio-demographic status and mental illnesses. There were 11 questions designed for adolescents, which formed the Adolescent Information Form, and there were 13 questions in the Parent Information Form [18,21,24–27]. The internalized stigmatization levels of the adolescents were determined by the Internalized Stigmatization of Mental Illness Scale—Adolescents Form (ISMI-AF). The Parent's Internalized Stigmatization of Mental Illness Scale (PISMI) was used to determine parents' internalized stigmatization levels.

**Internalized Stigma of Mental Illness Scale—Adolescents Form (ISMI-AF):** This scale was developed by Boyd-Ritsher et al. [28]. Its Turkish validity and reliability study on adolescents with mental illnesses was performed by Dikeç et al. [25]. It is a self-report scale consisting of 29 items, and it evaluates internalized stigma. It has five subscales: "alienation" (items: 1, 5, 8, 16, 17, 21), "stereotype endorsement" (items: 6, 10, 18, 19, 23, 29), "perceived discrimination" (items: 3, 15, 22, 25, 28), "social withdrawal" (items: 4, 9, 11, 12,

13, 20), and "stigma resistance" (items: 7, 14, 24, 26, 27). The scale is a four-point Likert-type scale; its response options are "strongly disagree" (1 point), "disagree" (2 points), "agree" (3 points), and "strongly agree" (4 points). Items in the "stigma resistance" subscale (7, 14, 24, 26, 27) are calculated inversely. The total ISMI-AF score ranges from one to five and is obtained by adding the scores of the scale items and dividing them by the number of items, with no cutoff score. Higher scores indicate that the internalized stigma of the adolescent is more severe. In the study of Dikeç et al. [25], the Cronbach's alpha coefficient was 0.84 for the total score, and, for the subscales, 0.78 for alienation, 0.67 for stereotype endorsement, 0.71 for perceived discrimination, 0.76 for social withdrawal, and 0.35 for stigma resistance. In the present study, the Cronbach's alpha coefficient was 0.87 for the total score, 0.78 for alienation, 0.61 for stereotype endorsement, 0.79 for perceived discrimination, 0.83 for social withdrawal, and 0.51 for stigma resistance.

**Parental Internalized Stigma of Mental Illness Scale (PISMI):** The Turkish validity and reliability study of the Parental Internalized Stigma of Mental Illness Scale, developed by Boyd-Ritsher et al. [28], was performed by Dikeç et al. [24]. The scale was obtained by adapting the statements in the ISMI to parents. Factor analysis was similar to that of the ISMI. The scale is a four-point Likert-type scale consisting of 29 items [18]. It has five subscales: "alienation" (items: 1, 5, 8, 16, 17, 21), "stereotype endorsement" (items: 6, 10, 18, 19, 23, 29), "perceived discrimination" (items: 3,15, 22, 25, 28), "social withdrawal" (items: 4, 9, 11, 12, 13, 20), and "stigma resistance" (items: 7, 14, 24, 26, 27). The total score is obtained by adding the scale scores and dividing them by the number of items, and there is no cutoff score for this scale. High scores indicate that the person's internalized stigma is more severe. In the study of Dikeç et al. [24], the Cronbach's alpha coefficient of the PISMI was 0.87 for the total score, and, for the subscales, 0.69 for alienation, 0.72 for stereotype endorsement, 0.76 for perceived discrimination, 0.76 for social withdrawal, and 0.81 for stigma resistance. In the present study, the Cronbach's alpha coefficient was 0.89 for the total score, 0.81 for alienation, 0.71 for stereotype endorsement, 0.78 for perceived discrimination, 0.84 for social withdrawal, and 0.41 for stigma resistance.

### 2.5. Data Analysis

The data of the present study were analyzed using SPSS 26.0, and the results were entered according to the APA (American Psychiatric Association) Publications and Communications Board Working Group on Journal Article Reporting Standards 8 [29]. Mean, standard deviation, minimum, maximum, total number, and percentage were used to analyze descriptive data. Kurtosis and Skewness analyzed the distribution of the mean scores of the total scale and subscales and determined that they had a normal distribution. Independent-t tests were used to compare the internalized stigma scale and subscale mean scores of adolescents and their parents. The Pearson correlation analysis was used to find the correlation between the two scales, and multiple linear regression analysis was used to evaluate the effect of the parents' PISMI total and subscale scores on adolescents' ISMI-AF total scores. The Cronbach's alpha coefficient was calculated in the PISMI and ISMI-AF reliability analyses. All findings were evaluated at a $p < 0.05$ significance level.

### 2.6. Ethical Considerations

Ethics committee permission was obtained from the Scientific Research Ethics Committee of the University of Health Sciences, dated 9 July 2020 and numbered E.22007. In addition, written consent was obtained from both the adolescents and their parents after obtaining institutional permission. Parents signed the informed consent form for themselves and their children in order to participate in the study.

### 3. Results

The mean age of the adolescents who participated in the study was 14.01 (1.79) and the mean age of their parents was 39.55 (11.35). Of the adolescents, 72% were male and 59.8% were secondary school students. Of the parents, 89.7% were married, 74.8% were

women, and 28% were illiterate. A total of 89.7% of adolescents and 92.5% of parents had social security; 57% of the adolescents and 67.3% of the parents stated that they perceived their economic status as moderate (Table 1). Of the adolescents, 4.7% (n = 5) had a history of psychiatry clinic visits and 84.1% (n = 90) of them used psychotropic medication. The average duration of outpatient follow-up of the adolescents was 35.60 (34.40) months, and the average duration of medication use was 35.31 (34.54) months.

**Table 1.** Socio-demographic characteristics of the participants.

|  | Adolescents | Parents |
|---|---|---|
| **Characteristics** | **n (%)** | **n (%)** |
| **Gender** |  |  |
| Female | 30 (28) | 80 (74.8) |
| Male | 77 (72) | 27 (16.7) |
| **Education status** |  |  |
| Illiterate | - | 30 (28) |
| Literate | - | 31 (29) |
| Primary school | 38 (35.5) | 1 (0.9) |
| Middle school | 64 (59.8) | 1 (0.9) |
| High school | 5 (4.7) | 21 (19.6) |
| University | - | 23 (21.5) |
| **Social insurance** |  |  |
| Yes | 96 (89.7) | 99 (92.5) |
| No | 11 (10.3) | 8 (7.5) |
| **Economic status** |  |  |
| Low | 9 (8.4) | 8 (7.5) |
| Moderate | 61 (57) | 72 (67.3) |
| High | 37 (34.6) | 27 (25.2) |
| **Employment status** |  |  |
| Employed | 3 (2.8) | 57 (53.3) |
| Unemployed | 104 (97.2) | 50 (46.7) |

While there was no significant difference between the total scores of internalized stigma and the subscale mean scores of the adolescents and their parents ($p > 0.05$), the subscale scores of stereotype endorsement were found to be significantly different ($p < 0.05$) (Table 2). In other words, the stigmatization levels of the adolescents and their parents were similar.

**Table 2.** Comparison of ISMI-AF and PISMI total scores and subscale scores.

|  | ISMI-AF | | PISMI | |  |  |
|---|---|---|---|---|---|---|
| **Scales** | **Min–Max** | **Mean (SD)** | **Min–Max** | **Mean (SD)** | **Test** | ***p*** |
| **Total score** | 1–2.72 | 1.72 (0.42) | 1.10–2.93 | 1.76 (0.41) | −0.67 | 0.25 |
| Alienation | 1–3.5 | 1.54 (0.66) | 1–3.33 | 1.57 (0.57) | −0.25 | 0.79 |
| Perceived discrimination | 1–3.6 | 1.61 (0.67) | 1–3.20 | 1.57(0.57) | 0.48 | 0.63 |
| Stereotype endorsement | 1–3 | 1.42 (0.44) | 1–3 | 1.58 (0.49) | **2.42** | **0.01** |
| Social withdrawal | 1–3.83 | 1.53 (0.61) | 1–3.17 | 1.56 (0.58) | −0.39 | 0.69 |
| Stigma resistance | 1–4 | 2.65 (0.63) | 1–3.8 | 2.63 (0.56) | 0.25 | 0.80 |

Independent-*t* testCorrelation analysis revealed a significant correlation at the $p < 0.05$ level between PISMI and ISMI-AF stereotype endorsement (r = 0.19, $p = 0.04$). There was a significant positive correlation between PISMI stereotype endorsement and ISMI-AF total and subscale scores. Additionally, correlation analysis revealed a significant positive correlation level between PISMI perceived discrimination and ISMI-AF total and stereotype endorsement subscale scores and stigma resistance subscales (Table 3).

**Table 3.** Correlation between ISMI-AF and PISMI total and subscale scores.

| | PISMI Total Score | Alienation | Perceived Discrimination | Stereotype Endorsement | Social Withdrawal | Stigma Resistance |
|---|---|---|---|---|---|---|
| **ISMI-AF total score** | **r: 0.19 \*** **p: 0.04** | r: 0.11 p: 0.22 | **r: 0.19\*** **p: 0.04** | **r: 0.27 \*\*** **p < 0.001** | r: 0.18 p: 0.06 | r: −0.39 p: 0.69 |
| Alienation | r: 0.11 p: 0.23 | r: 0.08 p: 0.40 | r: 0.11 p: 0.25 | **r: 0.21 \*** **p: 0.02** | r: 0.16 p: 0.09 | r: −0.15 p: 0.10 |
| Perceived discrimination | r: 0.15 p: 0.10 | r: 0.74 p:0.45 | r: 0.14 p:0.13 | **r: 0.24 \*** **p: 0.01** | r: 0.16 p:0.09 | r: −0.32 p: 0.74 |
| Stereotype endorsement | **r: 0.19 \*** **p: 0.04** | r: 0.12 p:0.18 | **r: 0.23 \*** **p: 0.01** | **r: 0.32 \*\*** **p <0.001** | r: 0.18 p: 0.06 | r: −0.11 p: 0.25 |
| Social withdrawal | r: 0.17 p: 0.07 | r: 0.11 p: 0.24 | r: 0.13 p: 0.15 | **r: 0.24 \*** **p: 0.01** | r: 0.15 p: 0.10 | r: −0.01 p: 0.85 |
| Stigma resistance | r: 0.09 p: 0.31 | r: 0.05 p: 0.56 | r: 0.09 p: 0.32 | r: 0.00 p:0.96 | r: 0.00 p: 0.97 | **r: 0.22** **p: 0.02** |

\*\* The Pearson L. Correlation is significant at the 0.01 level. \* Correlation is significant at the 0.05 level.

Multiple linear regression analysis was performed. The model, established with PISMI totals and subscales as well as those of the ISMI-AF, was significant, and it was determined that the stereotype endorsement subscale mean scores of the PISMI explained 7.4% of the ISMI-AF total score (Table 4) (Adjusted R square = 0.042).

**Table 4.** The effect of PISMI totals and subscales on the ISMI-AF.

| Dependent Variable | Independent Variables | B | ß | t | p | F | Model (p) | $R^2$ |
|---|---|---|---|---|---|---|---|---|
| **ISMI-AF** | Constant | 1.34 | | 5.29 | <0.001 | 1.78 | 0.11 | 0.096 |
| | PISMI | −0.58 | −0.57 | −0.32 | 0.74 | | | |
| | Alienation | −0.01 | −0.13 | −0.23 | 0.81 | | | |
| | Perceived discrimination | 0.03 | 0.22 | 0.52 | 0.60 | | | |
| | Stereotype endorsement | 0.07 | 0.50 | 1.05 | 0.29 | | | |
| | Social withdrawal | 0.03 | 0.26 | 0.46 | 0.64 | | | |
| | Stigma resistance | 0.03 | 0.15 | 0.36 | 0.71 | | | |
| | Constant | 1.35 | | 10.16 | <0.001 | 8.35 | <0.001 | 0.074 |
| | Stereotype endorsement | 0.03 | 0.27 | 2.89 | 0.05 | | | |

## 4. Discussion

This study aimed to evaluate and compare the internalized stigma of mental illnesses among adolescents with ADHD and their parents. No significant difference was found between their total scores of internalized stigma or their subscales mean scores, other than the subscale scores for stereotype endorsement, and it was concluded that the PISMI had a significant effect on the ISMI-AF. Dikeç et al. [21] conducted a study in an adolescent clinic for mental illnesses in Turkey and found no significant difference in internalized stigma between children and parents. The internalized stigmatization levels of participants in the current study were similar to those observed by Dikeç et al. [21].

The level of stereotype endorsement of the parents was higher than that of the adolescents in the present study. It might be that the parents were more aware of public stigmatization, stereotyping, and prejudices than the adolescents, or that they agreed

more with these stereotypes about their children [5]. A specific cognitive capacity must be matured in an individual to notice, perceive, and internalize stigma. In childhood, this cognitive capacity is not fully developed. This can be attributed to the cognitive capacity of adolescents [30]. On the other hand, parents might be affected by the condition of their children and their children's internalized stigmatization. Because of the educational, emotional, and social problems of their children, parents may be concerned about their social labels, isolation, and rejection, and worry about the potential effects of disorders and treatments, and about opportunities for their future plans [5]. Just as parents can be affected by all these factors, they can also affect their children due to their interaction with their children. If parents show negative parental behaviors such as criticism, disparagement, and irritability, these affect their interactions with their children negatively. In other words, internalized stigmatization leads to reduced social functionality and social skills, as well as aggressive behaviors in children. Future studies should determine the potential causes of internalized stigmatization within the two groups.

Mental health professionals should deeply explore parents' and adolescents' perceptions and attitudes, listen to their concerns, and train them by explaining the nature of ADHD, and thereby help to correct stereotypes and prejudices. They should use stigma-reducing interventions. It is important to understand which interventions can increase self-efficacy and decrease internalized stigmatization in children with ADHD and their parents. For example, in the ADHD field, cognitive and behavioral therapy proved to have beneficial effects on parental expectations, psychological well-being, and attribution of children's disruptive behaviors [31]. Wong et al. [31] found that the experimental group who received CBT experienced small to moderate effects on parenting distress and dysfunctional attitudes. Thus, decreased parental stress and internalized stigmatization are reflected in adolescents' improved self-esteem; hence, their educational, emotional, and social outcomes may improve, resulting in success in their academic pursuits, school life, and peer relations.

*Limitations of the Study*

The data from this study are limited to adolescents and their parents due to its cross-sectional design, which does not allow causal inferences. Further research with a longitudinal design could be conducted to examine the factors affecting the internalized stigmatization of both adolescents with ADHD and their parents.

**5. Conclusions**

In this study, there was no significant difference between the internalized stigma scores of the adolescents diagnosed with ADHD and those of their parents. There was a positive correlation between the two groups' internalized stigmatization scores. The PISMI scores affected the ISMI-AF scores, which may be interpreted as the parents' and adolescents' perspectives and attitudes toward stigmatization mutually affecting one another. For ADHD, whose frequency is increasing day by day, intervention studies should be conducted to reduce adolescents' and parents' internalized stigma.

**Author Contributions:** Conceptualization, G.D. and Ö.B.; methodology, G.D. and Ö.B.; formal analysis, G.D.; data collection and curation, C.K. and Ş.Y.S.; writing—original draft preparation, G.D.; writing—review and editing, G.D., Ö.B., C.K. and Ş.Y.S. All authors have read and agreed to the published version of the manuscript.

**Funding:** This research received no external funding.

**Institutional Review Board Statement:** The study was conducted according to the guidelines of the Declaration of Helsinki and was approved by the Hamidiye Scientific Ethics Committee of the University of Health Sciences (dated 9 July 2020 and numbered E.22007.). An institutional permit was obtained from the Celal Bayar University head physician.

**Informed Consent Statement:** Written consent for participating in the study was obtained from the adolescents and their legal guardians and parents.

**Data Availability Statement:** The data presented in this study are available on request from the corresponding author.

**Conflicts of Interest:** The authors declare no conflict of interest.

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
