# Peer review of "Do We Learn to Internalize Stigma from Our Parents? Comparison of Internalized Stigmatization in Adolescents Diagnosed with ADHD and Their Parents"

_adolescents, doi:10.3390/adolescents2040034_

Round 1

Reviewer 1 Report

GENERAL COMMENTS

The aim of this paper was to examine the relationship between internalized stigmatization levels in adolescents with ADHD and their parents. Although this article addresses an interesting topic, many issues should be addressed before publication. I suggest sending the manuscript for native English proofreading.

SPECIFIC COMMENTS

INTRODUCTION
The introduction needs major revision and clarification. First, the aim of the study is not clear. The way the authors build up their introduction does not lead to the research question. Although much of the necessary information regarding the background is already written down, the authors should re-structure their introduction, explaining why their research is important. The authors described some studies on ADHD and stigmatization separately, but the discussion of both together is very shallow. Thus, it is recommended that the authors expand this part: how adolescents with ADHD could lead to stigmatization among adolescent and their parents. More importantly, this should lead to a clear research question.

METHODS
The methods section needs major revision. As it stands, it is not possible to replicate the study.

Settings – Data was collected between July 2020 to March 2021, which is during the COVID-19 pandemic. Is the data reliable? No mention about COVID-19 in this manuscript?

Why sample calculation was not used? Seemed like from the results, the sample size is insufficient. Please justify.

Why need to bold the Parental Internalized stigma of Mental Illness Scale?

The procedure to collect the data was unclear.

RESULTS
I think the authors put too much information in the tables, making the entire manuscript hard to follow. Table 1 checklist needs more clarification – employment, participant in 12-18, already 3 participants working?

DISCUSSION
In the discussion section, the authors should further discuss their findings and the implication of these findings. They should also discuss their findings in more depth. However, in this section, the authors present many new statements, which should be moved to the introduction section. The authors also discuss many topics that are not related to the results. In addition, they describe many studies in great detail, which is not necessary for the discussion. This makes the discussion not organised and difficult to follow. The limitation is also not well thought. Please revise.

Thank you.

Author Response

Reviewer I Comments

Authors’ Response

GENERAL COMMENTS

The aim of this paper was to examine the relationship between internalized stigmatization levels in adolescents with ADHD and their parents. Although this article addresses an interesting topic, many issues should be addressed before publication. I suggest sending the manuscript for native English proofreading.

Thank you for your contributions. The manuscript has edited by native speaker. The certificate was attached.

INTRODUCTION
The introduction needs major revision and clarification.

First, the aim of the study is not clear.

The way the authors build up their introduction does not lead to the research question.

Although much of the necessary information regarding the background is already written down, the authors should re-structure their introduction, explaining why their research is important. The authors described some studies on ADHD and stigmatization separately, but the discussion of both together is very shallow.

Thus, it is recommended that the authors expand this part: how adolescents with ADHD could lead to stigmatization among adolescent and their parents.

More importantly, this should lead to a clear research question.

Thank you for your recommendation.

We rebuilt the introduction part and the added some information internalized stigmatization among both adolescents with ADHD and their parents (lines 55-95).

We removed some information from discussion to introduction to clarify the research questions (lines 55-95).

METHODS
The methods section needs major revision. As it stands, it is not possible to replicate the study.

Settings – Data was collected between July 2020 to March 2021, which is during the COVID-19 pandemic. Is the data reliable? No mention about COVID-19 in this manuscript?

Why sample calculation was not used? Seemed like from the results, the sample size is insufficient. Please justify.

Why need to bold the Parental Internalized stigma of Mental Illness Scale?

The procedure to collect the data was unclear.

Thank you for your contributions.

We add new subtitle data collection procedure to explain why sample calculation was not used, COVID-19 pandemic and how it affected the data collection (lines 109-119).

RESULTS
I think the authors put too much information in the tables, making the entire manuscript hard to follow. Table 1 checklist needs more clarification – employment, participant in 12-18, already 3 participants working?

Thank you for you recommendation, we tried to write only major of the socio-demographic data of adolescents and their parents (line 212-216) . Yes, it is correct, 3 of participants between 12-18 was employment.

DISCUSSION
In the discussion section, the authors should further discuss their findings and the implication of these findings. They should also discuss their findings in more depth. However, in this section, the authors present many new statements, which should be moved to the introduction section. The authors also discuss many topics that are not related to the results. In addition, they describe many studies in great detail, which is not necessary for the discussion. This makes the discussion not organised and difficult to follow. The limitation is also not well thought. Please revise.

We reorganized the discussion part according to results of this study deeply (lines 273-299).

We added potential limitation on the limitation section (lines 303-306).

Reviewer 2 Report

This is a review of the manuscript titled "Do We Learn to Internalize Stigma from Our Parents? Comparison of Internalized Stigmatization of Adolescents Diagnosed With ADHD and Their Parents". The aim of this study is to compare the internalized stigma of adolescents with ADHD and their parents. It was found that ADHD adolescents reported a significantly lower stereotype-endorsement subscale score compared with their parents. The total internalized stigma score and the other subscale scores did not differ between ADHD adolescents and their parents. ADHD adolescents’ internalized stigma was positively associated with their parents’ internalized stigma. This manuscript could be improved if the following concerns are addressed:

1. I suggest the authors introduce the diagnostic criteria for ADHD in the DSM-V.

2. Please explain in more detail why it is theoretically interesting to examine the differences in internalized stigma between adolescents with ADHD and their parents.

3. Please provide directional hypotheses. Are ADHD adolescents expected to have higher or lower levels of internalized stigma in comparison with their parents? Please provide explanations as well.

4. It is stated that, "Only one parent of each adolescent was included in the study" (p. 3). How did the author determine which parent was included in the study?

5. Please include sample items for each subscale.

6. The authors reported the results of the factor analysis of the Parental Internalized Stigma of Mental Illness Scale (PISMI). Please clarify whether the factor analysis was performed using the present data or conducted in a previous study.

7. Please clarify what kind of t-tests were conducted. Were paired samples t-tests used?

8. The authors only examined the association between ADHD adolescents’ overall internalized and parents’ overall internalized stigma using correlation and regression analyses. I recommend the authors examine the correlations for the subscale scores as well.

9. On the basis of the correlation and regression analysis, the authors suggests that ADHD adolescents’ internalized stigma is affected by parents’ internalized stigma. Please discuss whether it is possible that parents’ internalized stigma is also affected by ADHD adolescents’ internalized stigma.

10. It is stated that intervention studies are recommended. I suggest the authors specify what types of interventions should be used for ADHD adolescents and their parents. For example, past research has supported the effectiveness of cognitive-behavior therapy (CBT) for parents of ADHD children (Wong et al., 2018).

Reference

Wong, D. F., Ng, T. K., Ip, P. S., Chung, M. L., & Choi, J. (2018). Evaluating the effectiveness of a group CBT for parents of ADHD children. Journal of Child and Family Studies, 27, 227-239. https://doi.org/10.1007/s10826-017-0868-4

11. This study used a cross-sectional design, which does not allow causal inferences. Future research with a longitudinal design is recommended. This is an additional limitation of this study.

Author Response

Reviewer II Comments

Authors’ Response

This is a review of the manuscript titled "Do We Learn to Internalize Stigma from Our Parents? Comparison of Internalized Stigmatization of Adolescents Diagnosed with ADHD and Their Parents". The aim of this study is to compare the internalized stigma of adolescents with ADHD and their parents. It was found that ADHD adolescents reported a significantly lower stereotype-endorsement subscale score compared with their parents. The total internalized stigma score and the other subscale scores did not differ between ADHD adolescents and their parents. ADHD adolescents’ internalized stigma was positively associated with their parents’ internalized stigma. This manuscript could be improved if the following concerns are addressed:

Thank you for your contributions.

1. I suggest the authors introduce the diagnostic criteria for ADHD in the DSM-V.

We added diagnostic criteria of DSM-V for ADHD at the beginning of the introduction part (lines 30-33).

2. Please explain in more detail why it is theoretically interesting to examine the differences in internalized stigma between adolescents with ADHD and their parents.

We tried to explained the problems and why is worth to examine at the introduction part (lines 61-91).

3. Please provide directional hypotheses. Are ADHD adolescents expected to have higher or lower levels of internalized stigma in comparison with their parents? Please provide explanations as well.

4. It is stated that, "Only one parent of each adolescent was included in the study" (p. 3). How did the author determine which parent was included in the study?

We explained this statement at the data collection section (lines 109-119).

5. Please include sample items for each subscale.

We added items for each subscale to PISMI (lines 168-172).

6. The authors reported the results of the factor analysis of the Parental Internalized Stigma of Mental Illness Scale (PISMI). Please clarify whether the factor analysis was performed using the present data or conducted in a previous study.

It was conducted in a previous validity and reliability study. We added this explanation in line 168.

7. Please clarify what kind of t-tests were conducted. Were paired samples t-tests used?

We added Independent t-test at the data analysis section (line 192).

8. The authors only examined the association between ADHD adolescents’ overall internalized and parents’ overall internalized stigma using correlation and regression analyses. I recommend the authors examine the correlations for the subscale scores as well.

We reperformed the correlation analysis with subscales scores (line 187-188 and Table 3).

9. On the basis of the correlation and regression analysis, the authors suggests that ADHD adolescents’ internalized stigma is affected by parents’ internalized stigma. Please discuss whether it is possible that parents’ internalized stigma is also affected by ADHD adolescents’ internalized stigma.

We added this statement at discussion section (lines 278-288).

10. It is stated that intervention studies are recommended. I suggest the authors specify what types of interventions should be used for ADHD adolescents and their parents. For example, past research has supported the effectiveness of cognitive-behavior therapy (CBT) for parents of ADHD children (Wong et al., 2018).

Reference

Wong, D. F., Ng, T. K., Ip, P. S., Chung, M. L., & Choi, J. (2018). Evaluating the effectiveness of a group CBT for parents of ADHD children. Journal of Child and Family Studies, 27, 227-239. https://doi.org/10.1007/s10826-017-0868-4

Thank you for recommendation. The reference was cited as explaining the specific interventions to enhance self-efficiency of both parents and adolescents with ADHD (lines 294-298).

11. This study used a cross-sectional design, which does not allow causal inferences. Future research with a longitudinal design is recommended. This is an additional limitation of this study.

Thank you for your contributions. We added this limitation and recommendation (lines 303-306)

Round 2

Reviewer 2 Report

This is a review of the revised version of the manuscript titled "Do We Learn to Internalize Stigma from Our Parents? Comparison of Internalized Stigmatization of Adolescents Diagnosed With ADHD and Their Parents". This manuscript has been improved. However, there are remaining issues to be addressed:

1. Please provide directional hypotheses. Are ADHD adolescents expected to have higher or lower levels of internalized stigma in comparison with their parents? Please provide explanations as well.

2. As this study used 107 adolescent-parent dyads, it is not appropriate to use independent t-tests to examine the differences between adolescents and their parents. Please reanalyze the data using paired samples t-tests.

3. The authors only used ADHD adolescents’ overall internalized and parents’ overall internalized stigma in the regression analysis. The result is the same as the correlation between the two scores. I recommend the authors used the subscale scores in the regression analysis

Author Response

Reviewer II Comments

Authors’ Response

This is a review of the revised version of the manuscript titled "Do We Learn to Internalize Stigma from Our Parents? Comparison of Internalized Stigmatization of Adolescents Diagnosed with ADHD and Their Parents". This manuscript has been improved. However, there are remaining issues to be addressed:

The manuscript was improved with your contributions. Thank you for your contributions.

1. Please provide directional hypotheses. Are ADHD adolescents expected to have higher or lower levels of internalized stigma in comparison with their parents? Please provide explanations as well.

Since the design of this study is descriptive and to our best knowledge, there was no study which is evaluated the internalized level of both adolescents with ADHD and their parent, the study has the research question. Because research questions outline the phenomena under study, who were studied, and what the authors wanted to know about them. A question is used when we do not have a particular hunch or hypothesis about the outcome of the study. Therefore, we did not add any hypotheses (Connelly, 2015).

Connelly, L. M. (2015). Research questions and hypotheses. Medsurg Nursing24(6), 435-436.

2. As this study used 107 adolescent-parent dyads, it is not appropriate to use independent t-tests to examine the differences between adolescents and their parents. Please reanalyze the data using paired samples t-tests.

As the adolescents and their parents are different/independent two sample, Independent t-test was used. Dependent (related, within subject or paired) sample t-test compares the means of two conditions in which the same participants participated in the study. Independent sample t-test compares the means of two groups of participants (Rochon et al., 2012; Gerald, 2018).

Gerald, B. (2018). A brief review of independent, dependent and one sample t-test. International Journal of Applied Mathematics and Theoretical Physics4(2), 50-54.

Rochon, J., Gondan, M., & Kieser, M. (2012). To test or not to test: Preliminary assessment of normality when comparing two independent samples. BMC Medical Research Methodology12(1), 1-11.

3. The authors only used ADHD adolescents’ overall internalized and parents’ overall internalized stigma in the regression analysis. The result is the same as the correlation between the two scores. I recommend the authors used the subscale scores in the regression analysis

Thank you for your recommendation. We have added the subscales scores in regression analysis and written lines 251-155 and Table 4.